# Framework for Generation and Removal of Multiple Types of Adverse Weather from Driving Scene Images

**DOI:** 10.3390/s23031548

**Published:** 2023-01-31

**Authors:** Hanting Yang, Alexander Carballo, Yuxiao Zhang, Kazuya Takeda

**Affiliations:** 1Graduate School of Informatics, Nagoya University, Furo-cho, Chikusa-ku, Nagoya 464-8601, Japan; 2Faculty of Engineering and Graduate School of Engineering, Gifu University, 1-1 Yanagido, Gifu City 501-1193, Japan; 3Institute of Innovation for Future Society, Nagoya University, Furo-cho, Chikusa-ku, Nagoya 464-8601, Japan; 4Tier IV Inc., Nagoya University Open Innovation Center, 1-3, Mei-eki 1-chome, Nakamura-Ward, Nagoya 450-6610, Japan

**Keywords:** Multiple Weather Translation, unpaired training, CycleGAN

## Abstract

Weather variation in the distribution of image data can cause a decline in the performance of existing visual algorithms during evaluation. Adding additional samples of target domain to training data or using pre-trained image restoration methods such as de-hazing, de-raining, and de-snowing, to improve the quality of input images are two promising solutions. In this work, we propose Multiple Weather Translation GAN (MWTG), a CycleGAN-based, dual-purpose framework that simultaneously learns weather generation and its removal from image data. MWTG consists of four GANs constrained using cycle consistency that carry out domain translation tasks between hazy, rainy, snowy, and clear weather, using an asymmetric approach. To increase network capacity, we employ a spatial feature transform (SFT) layer to fuse the features extracted from the weather layer, which contains high-level domain information from the previous generators. Further, we collect an unpaired, real-world driving dataset recorded under various weather conditions called Realistic Driving Scenes under Bad Weather (RDSBW). We qualitatively and quantitatively evaluate MWTG using the RDSBW and the variation of Cityscapes that synthesize weather effects, eg., FoggyCityscape. Our experimental results suggest that MWTG can generate realistic weather in clear images and also accurately remove noise from weather images. Furthermore, the SOTA pedestrian detector ASCP is shown to achieve an impressive gain in detection precision after image restoration using the proposed MWTG method.

## 1. Introduction

Autonomous vehicles are highly dependent on their perception systems to sense the surrounding environment and detect other road users, as this information is essential for their planning and control systems. However, when adverse weather occurs, retrieved images with low contrast and poor visibility can degrade the performance of the visual algorithms used in autonomous vehicle perception systems, such as detection, tracking, and intention estimation [1,2,3,4,5]. This degradation is caused by tiny particles in the atmosphere that absorb and reflect light [6]. To improve the performance of perception systems during adverse weather, researchers have used de-hazing, de-raining, and de-snowing applications to remove these effects to obtain clearer images.

Previous works have suggested using heuristic priors based on observations to remove weather-related degradation from images, such as Dark Channel Prior [7], Haze-Lines Prior [8], Sparse Coding [9], Layer Prior [10], Saturation and Visibility Features [11], and Mutiple-guided Filter [12]. These works utilize meteorological models, with an example of a haze model being presented in Equation (Equation 1), where I is the input hazy image, J is its radiance, A is the atmospheric light, and *t* is the transmittance map measuring the amount of light able to pass through the haze layer [13].
(1)I(x)=J(x)t(x)+A(1−t(x))

The same paradigm can also be used for de-raining and de-snowing images. The simplified models for simulating rain [14] and snow [15] are shown in Equations (Equation 2) and (Equation 3), respectively.
(2)O(x)=B(x)+R(x)
(3)I(x)=K(x)t(x)+A(1−t(x))
where O(·), B(·), and R(·) symbolize a rainy image, a clean background, and a rain layer, respectively. Equation (Equation 3) utilizes K(·) to represent a snow image and the concept is similar to that explained in Equation (Equation 1) and (Equation 2).

With the advancement of deep learning techniques, researchers have explored alternative solutions to mitigate the limitations of traditional approaches in generalization ability. One such approach that has gained traction is convolutional neural networks (CNN) [16,17,18,19,20,21] and generative adversarial networks (GAN) [22,23] trained on prepared datasets. These methods allow for direct mapping of the domain variation between paired weather-affected and clear images.

However, there are several reasons for investing in a new approach. First of all, most current weather removal methods are designed to only solve one type of weather problem. Second, researchers still tend to utilize paired datasets containing synthetic weather images, even though using unpaired, real-world weather images would likely provide more authentic and richer details for deep modeling. Third, current restoration models usually only allow one-way processing for the removal of weather data, although studies have shown that models that learn generation and removal simultaneously can achieve better transformation results. Finally, generating inclement weather images and using them to train visual algorithms to output clear images could improve autonomous vehicle sensing under adverse weather conditions. Therefore, in this study, we proposed a dual-purpose framework for generating images of multiple adverse weather conditions from clear weather images, as well as the removal of these conditions. Our contributions are as follows:A deep-learning model for the generation and removal of multiple weather conditions in visual images. Inspired by CycleGAN, we introduce a model consisting of four generators and four discriminators. The four GANs are trained to generate and remove haze, rain, snow, and clear weather conditions. Once training is completed, the clear weather generator can be used to restore images, no matter which of the three weather conditions described above is present.A disentangled training strategy based on unpaired, real-world weather images. By training our model with images of real weather conditions, ground truth images and parameters for meteorological physical models are unnecessary.We have meticulously curated our newly created dataset, Realistic Driving Scene under Bad Weather (RDSBW) [24], by removing images with blur and wiper. The dataset now comprises of 20,000 1080P images extracted from videos. These images showcase driving scenes captured through the windshield in varying weather conditions such as clear, hazy, and snowy.Evaluate the performance of our model using the RDSBW and Weather Cityscapes datasets. Use structural similarity (SSIM) and peak signal-to-noise ratio (PSNR) to measure image quality before and after removing adverse weather. Furthermore, assess detection accuracy using a state-of-the-art pedestrian detector, demonstrating promising results for both accuracy and speed.

## 2. Related Work

### 2.1. Removal of One Type of Adverse Weather

Based on image restoration methods, approaches for removing a single type of adverse weather gained extensive attention due to their possible applications in the field of computer vision. As a result, many researchers have proposed methods of de-hazing, de-raining, and de-snowing images.

#### 2.1.1. De-Hazing

Early on, learning-based de-hazing models required an intermediate step to estimate the transmission map and atmospheric light [25,26,27,28]. Error or bias during this estimation process resulted in artifacts or color distortion, requiring extra processing to reconstruct clear images [29]. In contrast, more recent, deep-learning models map the relationships between hazy and clear images directly using datasets of paired images with ground truth labels  [16]. Researchers have also proposed various, innovative modules to enhance their de-hazing networks, resulting in substantial advances. These include the feature attention module [17], dense feature fusion module [18], spatially-weighted channel attention module [19], etc.

#### 2.1.2. De-Raining

Deep learning methods for de-raining were first introduced in 2017, when Yang et al. [20] proposed region-dependent rain models with contextualized dilated modules to jointly detect and remove rain. In the same year, Fu et al. [21] proposed De-rainNet, which only uses high-frequency details as inputs. These two pioneering studies inspired other CNN-based methods that imported more advanced network architecture designs, achieving better results. However, using synthetic images of rainy weather as supervision often produces unsatisfactory results when processing images of real-world rainy weather.

The most potent generative models proposed to date incorporate generative adversarial networks (GAN) and exploit the Min–Max algorithm to reduce the distance between two domains, which can be used to bridge the visual gap between synthetic and real rain [14]. Zhang et al. [22] have suggested an effective, conditional, GAN-based single image de-raining framework with a novel, densely-connected generator and multi-scale discriminator. Li et al. [23] have proposed a coarse-to-fine framework that first calculates the underlying physics and then recovers the background using a depth-guided GAN.

#### 2.1.3. De-Snowing

Compared to de-hazing and de-raining, learning-based de-snowing is still in the early stages of its development due to variations in the shapes and sizes of snowflakes. Liu et al. [30] created multi-stage DesnowNet, a network that consists of translucency recovery modules and residual generation modules. Thanks to its context-aware features and loss functions, DesnowNet outputs images with better illumination, color, and contrast distributions than traditional methods.

To resolve problems caused by snowflake diversity, Chen et al. [15] proposed a hierarchical network, optimized using contradictory channel loss, that was based on contradicting the state of the prior channel. Zhang et al. [31] proposed using a framework with four sub-networks, including their newly-designed DDMSNet, in order to utilize semantic and depth information. After learning the semantic and geometric representations of snowy images, it was possible to recover clean images with clearer details.

### 2.2. Removal of Multiple Types of Adverse Weather

Only a few studies have been published which propose generic solutions for removing multiple types of adverse weather interference from visual images. Li et al. [32] used the Network Architecture Search framework to obtain a network that uses an image degraded by any type of weather condition as the input and predicts a clean image as the output. This All-in-One method was tested across three datasets of rainy, hazy, and snowy images and achieved better or comparable performance than dedicated adverse weather removal models. Jeya et al. [33] proposed a transformer-based encoder–decoder network called TransWeather. Through fine filtering, they created a dataset combining the Snow100K, Raindrop, and Outdoor-Rain corpora. Extensive experiments using multiple synthetic and real-world images proved that TransWeather can effectively remove any type of weather degradation present in an image.

### 2.3. Image-to-Image Translation and CycleGAN

If we think of adverse and clear weather as two different domains, the removal of weather effects can be viewed as a type of image-to-image translation process, the objective of which is transforming an image from one domain to the other [34]. The early work on image-to-image translation applied conditional adversarial networks to learn the mapping between the domains, which required paired datasets. This approach is not suitable for weather removal tasks, however, because we cannot collect data on different weather conditions while keeping the background the same at the pixel level, since atmospheric lighting is constantly changing. However, unpaired image-to-image translation is a possible solution. In order to preserve the attributes of the source image, as well as the relationships between objects, pioneering methods such as CycleGAN [35], DiscoGAN [36], and DualGAN [37] have employed GAN-based reconstruction objective functions, such as the one shown in Equation (Equation 4):(4)$$\begin{array}{c} L_{r}=∥G_{AB}\left(G_{BA}\left(x_{a}\right)\right)-x_{a}∥+∥G_{BA}\left(G_{AB}\left(x_{b}\right)\right)-x_{b}∥ \end{array}$$
where $L_{r}$ represents reconstruction loss, $G_{AB}$ represents the generator from domain *A* to domain *B*, and $G_{BA}$ is the generator from domain *B* to domain *A*, while $x_{a}$ and $x_{b}$ represent examples of the two domains.

The approach proposed in this paper is related to previous methods for removing degradation caused by multiple types of adverse weather, but it is not restricted to eliminating noise. In contrast, we use information obtained from the image translation domain and define the problem as unpaired translations of multiple types of weather, which can transform images from any weather domain into the clear domain, or vice versa, without providing a ground truth.

## 3. Multiple Weather Translation GAN

We propose a novel, multiple weather translation GAN called MWTG, inspired by CycleGAN [35]. The goal of our work is to translate clean images of traffic scenes into versions of these images with different types of weather degradation and to then be able to convert the weather-degraded images back into clean ones. The proposed method can also be used to convert real-world, weather-degraded images into clearer ones. Overall, MWTG consists of three GANs for weather effect generation and one GAN for weather effect removal. Our rationale for creating a multi-weather application was based on the observation that it would be convenient to be able to use just one model to remove all the various types of adverse weather effects that drivers are likely to encounter.

### 3.1. General Pipeline

To explain the theoretical basis for our approach in more detail, suppose haze, rain, and snow are three sub-domains of an adverse weather set ($X=x_{1},x_{2},x_{3}$) and that *Y* represents a clear weather domain. As shown in Figure 1, there exist three mappings from adverse to clear weather: $x_{1}\to Y$, $x_{2}\to Y$, and $x_{3}\to Y$. Furthermore, conversely, there also exist three mappings from clear to adverse weather: $Y\to x_{1}$, $Y\to x_{2}$, and $Y\to x_{3}$. In order to simplify the mapping process, we compress the mapping X→Y into one network. Therefore, our model requires four generators: $G_{A},G_{B},G_{B}$ for generating adverse weather effects (haze, rain, and snow, respectively) and $G_{D}$ for adverse weather removal. Correspondingly, four discriminators ($D_{A}$, $D_{B}$, $D_{C}$, and $D_{D}$) are introduced to distinguish real images from the generated, fake images. The pseudo-code of MWTG is also provided in Algorithm 1.
**Algorithm** **1** Multiple Weather Translation GAN (MWTG) **Input**: Training data pairs (A,B,C,D)    ▹ In order of haze, rain, snow, and clear **Output**: Generator networks $G_{A},G_{B},G_{C},G_{D}$1:Initialize generators and discriminators2:Define loss functions3:Define optimizers for generator and discriminator4:**while** 
epoch≤total_epoches 
**do**5:     **for** data pair (A, B, C, D) **in** data_loader **do**6:           Generate fake images: $F_{DA}=G_{D}\left(A\right)$, $F_{DB}=G_{D}\left(B\right)$, $F_{DC}=G_{D}\left(C\right)$ and $F_{A}=G_{A}\left(D\right)$, $F_{B}=G_{B}\left(D\right)$, $F_{C}=G_{C}\left(D\right)$7:           Generate reconstruct images: $R_{A}=G_{A}\left(F_{DA}\right)$, $R_{B}=G_{B}\left(F_{DB}\right)$, $R_{C}=G_{C}\left(F_{DC}\right)$ and $R_{DA}=G_{D}\left(F_{A}\right)$, $R_{DB}=G_{D}\left(F_{B}\right)$, $R_{DC}=G_{D}\left(F_{C}\right)$8:           **Update** Discriminator $D_{A}$, $D_{B}$, $D_{C}$ and $D_{D}$9:           **Update** Generator $G_{A}$, $G_{B}$, $G_{C}$ and $G_{D}$10:   **end for**11:**end while**

Since we want to translate the unpaired, real-world weather images, MWTG borrows the cycle consistency principle from pioneering works [35,38,39,40] to regulate the structure of the output images, so they remain the same as the input images. Therefore, after an image is translated by the weather removal/generation network, we can translate it back into its original domain using the same generator.

We are using A∼D to represent sets of hazy, rainy, snowy, and clear weather images, respectively. As part of a single processing step, the image data are simultaneously sorted into two different places. On the one hand, real *A*, real *B*, and real *C* are input to $G_{D}$ and the fake clear images are output. These fake images will then be input to $G_{A}∼G_{C}$ to obtain the reconstructed adverse weather images. On the other hand, real *D* images will be simultaneously input to $G_{A}∼G_{C}$ to obtain fake, adverse weather images. These results then go through $G_{D}$ to obtain the reconstructed clear images.

### 3.2. Weather Generators and Discriminators

As the backbones of our four MWTG generators, ResNet [41] (with a Residual Block) is used to maintain the previous output through a skip connection, a method which has been proven to be effective when training deeper neural networks. The input image will first be scaled down twice, using large convolutional filters. After obtaining the desired resolution, the first layer feature maps of the image will go through nine ResNet blocks, generating denser representations with more channels. In a similar manner to the encoder–decoder architecture used in [42], two transpose convolutional layers then follow, to reverse the dense representations back into normal size RGB images.

For the discriminators, we use simple, three-layer convolutional neural networks that gradually increase the number of filters. The last layer outputs a one-channel prediction map, which is the encoding input for the criterion function. Because our datasets consist of high-resolution images, it would be time and memory-consuming to infer the entire images. Therefore, in the training stage, we crop the images into 480×480 pixel patches to reduce the calculation burden, which are then learned using PatchGANs [35,43,44].

### 3.3. Weather Information Guidance

To obtain better results, we introduce a disentangled training strategy [24,45] that regards images degraded by adverse weather as composites of a weather layer and a clean background. We can then calculate the numerical distance between the input and output of each generator and store those distance values in a tensor that has the same dimensions as the input image. We refer to this tensor as the weather layer.

To provide additional input to the generator, we incorporate the spatial feature transform (SFT) to combine the weather layer feature with the extracted feature maps, allowing the weather layer to serve as guidance. SFT, which was first introduced by Wang et al. [46] for super-resolution image reconstruction, fuses the middle layer’s features with the image’s original features spatially using Affine transformations. We adopt the method proposed by Shao et al. [47] and use a two-layer convolution module to extract the condition map ϕ from the weather layer. The extracted map is then fed into the two convolutional layers to predict the modulation parameters γ and β. Lastly, we use Equation (Equation 5) to obtain the shifted features.

We then use the feature maps of the penultimate convolutional layer of the GAN generator as input *F* to the SFT module, while the fake image output from the SFT module is similar to the input image, the values of the elements in the weather layer are close to 0, which is the consequence of the vanishing gradients. That is why we normalize the weather layer before it reaches the SFT module:(5)SFT(F∣γ,β)=γ⊙F⊕β
where ⊙ means element-wise product and ⊕ means element-wise summation.

### 3.4. Loss Function

Three kinds of loss functions are used when formulating an MWTG model; adversarial loss, cycle consistency loss, and identity loss.

#### 3.4.1. Adversarial Loss

We use adversarial losses to obtain four mappings, three for from clear to adverse weather ($Y\to x_{1}$, $Y\to x_{2}$, $Y\to x_{3}$) and one for from adverse to clear weather (X→Y). The first three mappings can be expressed as shown in Equations (Equation 6):(6)$$\begin{array}{cc} \sum_{i=A,j=1}^{C,3}L_{GAN} & \left(G_{i},D_{i},Y,x_{j}\right)=E_{i∼p_{\mathrm{data}}\left(i\right)}\left[\log D_{i}\left(i\right)\right]+E_{D∼p_{\mathrm{data}}\left(D\right)}\left[1-\log D_{i}(G_{i}\left(D\right)\right] \end{array}$$
where $G_{A∼C}$ tries to generate images $G_{A∼C}\left(D\right)$ that look similar to images from domain $x_{1∼3}$, while $D_{A∼C}$ aims to distinguish between the translated samples $G_{A∼C}\left(D\right)$ and real samples *D*. The base of the logarithm in the equation is usually set to 2 or *e*.

The transformation from adverse to clear weather involves three components, corresponding to each weather sub-domain; the mean values of which are calculated as follows:(7)$$\begin{array}{cc} L_{GAN} & \left(G_{D},D_{D},X,Y\right)=\frac{1}{3}\sum_{i=1}^{n}L_{GAN}\left(G_{D},D_{D},x_{i},Y\right) \end{array}$$
where the $L_{GAN}$ over $x_{i}$ tries to enable $G_{D}$ to generate better haze, rain, and snow images, while $D_{D}$ needs to identify fake images after the generator is evolved.

#### 3.4.2. Cycle Consistency Loss

The concept of ‘cycle consistency loss’ was introduced in [35], the paper proposing CycleGAN. It is calculated as the L1 norm between the input image and the reconstructed image and is used to prevent the second generator from generating random images of the target domain. An example of forward cycle consistency is shown in Figure 2, where images of each type of adverse weather are first translated into the “clear” domain before being restored to the original adverse weather images. This process can be formulated as follows:$$A\to G_{D}\left(A\right)\to G_{A}\left(G_{D}\left(A\right)\right)\approx A,B\to G_{D}\left(B\right)\to G_{B}\left(G_{D}\left(B\right)\right)\approx B,\;\mathrm{and}\;C\to G_{D}\left(C\right)\to G_{C}\left(G_{D}\left(C\right)\right)\approx C.$$

Likewise, for backward cycle consistency, the clear image that is first translated into various weather domains should be restored to the same state as input, i.e.:$$D\to G_{A}\left(D\right)\to G_{D}\left(G_{A}\left(D\right)\right)\approx D,D\to G_{B}\left(D\right)\to G_{D}\left(G_{B}\left(D\right)\right)\approx D,\;\mathrm{and}\;D\to G_{C}\left(D\right)\to G_{D}\left(G_{C}\left(D\right)\right)\approx D.$$

To force the weather removal generator $G_{D}$ to update at the same pace as the adverse weather generators, we compute the average of the three cycle losses as the loss of $G_{D}$:(8)$$\begin{array}{cc} L_{cyc} & \left(G,D\right)=\sum_{i=A}^{C}E_{A∼p_{\mathrm{data}}\left(i\right)}\left[∥G_{i}{(G_{D}\left(i\right)-i∥}_{1}\right]+\frac{1}{3}\sum_{j=1}^{C}E_{D∼p_{\mathrm{data}\;}\left(D\right)}\left[∥G_{D}{(G_{j}\left(D\right)-D∥}_{1}\right] \end{array}$$

#### 3.4.3. Identity Loss

Identity loss is used to preserve the image color composition when applying painting transfer to realistic photo tasks. We also find it useful when dealing with large weather images that have obvious base color tones. The goal is to train the generator to learn to map the identities of the target domain images used as input. This identity loss can be expressed as:(9)$$\begin{array}{cc} L_{identity} & \left(G_{A},G_{B},G_{C},G_{D}\right)=\sum_{i=A}^{D}E_{i∼p_{\mathrm{data}\;}\left(i\right)}\left[∥G_{i}{\left(i\right)-i∥}_{1}\right] \end{array}$$

#### 3.4.4. Overall Objective Function

Based on the context provided above, the overall objective function can be formulated as shown in Equation (Equation 10), where $\lambda_{c}$ and $\lambda_{i}$ are weights that control the cycle consistency loss and identity loss:(10)$$\begin{array}{c} L_{obj}=L_{GAN}+\lambda_{c}L_{cyc}+\lambda_{i}L_{identity} \end{array}$$

## 4. Evaluation

In this section, we present the results of our qualitative and quantitative evaluations and also discuss the results when the mixed dataset or the new RDSBW dataset is used as an input.

### 4.1. Datasets

#### 4.1.1. Cityscapes Weather Datasets

Cityscapes [48] is an annotated corpus of 5000 driving scene images captured in urban areas. The researchers also simulated various weather effects onto the dataset, using information such as depth maps based on atmospheric scattering models. Foggy Cityscapes [49] includes three different haze densities for each image, representing visibilities of 150 m, 300 m, or 600 m, respectively. Rain Cityscapes [50] is based on 295 images, which were used to generate 36 different haze concentrations and rain types for each image. The training and testing sets of the Snow Cityscapes [31] each consist of 2000 pairs of images. The size of the images in both the training and testing sets is 512×256.

To train our MWTG, we reorganized the datasets described above to create a Cityscapes weather dataset, as shown in Figure 3. The synthetic weather datasets use the same depth maps as the background more than once to simulate different weather intensities. Since low-intensity weather does not degrade visual applications very much and high-intensity weather occurs relatively infrequently, we only used the 300-m haze images of the Foggy Cityscapes and chose 12 types of rain patterns from Rain Cityscapes. To keep all the images in the training set at the same resolution, which is important to reduce domain difference, we resized the Snow Cityscapes images to 2048×1024 using normal linear interpolation.

#### 4.1.2. Realistic Driving Scenes under Bad Weather Dataset

RDSBW is a new dataset we created, which includes 2831 clear, 2052 hazy, 4171 rainy, and 4777 snowy images. We picked out high-quality images, 1920×1080 pixels in size, from driving scene videos recorded in urban settings. Figure 4 shows examples of RDSBW images from each set.

### 4.2. Implementation Details

We used the Pytorch framework for training, testing, and image preprocessing. Two NVIDIA RTX A6000 graphics cards were used for training, with a batch size of 4. We trained the MWTG model for 200 epochs on each dataset to ensure convergence, using the Adam optimizer and a step learning rate schedule. In addition, we set the $\lambda_{c}$ and $\lambda_{i}$ loss weights at 10 and the extra identity loss weight $\lambda_{idt}$ at 2.

### 4.3. Evaluation Results and Comparisons

#### 4.3.1. Qualitative Evaluation Using RDSBW Data

We first conducted a qualitative experiment using MWTG with the RDSBW dataset. The samples of the weather generation results are shown in Figure 5. MWTG can translate an unseen clear image into hazy, rainy, and snowy images without changing the original background context. The proposed method seemed especially effective for adding haze and rain, based on the following three aspects. First, the color of the image shifted based on the type of adverse weather effect being added. Second, the weather effects were similar to those observed in real scenes, since MWTG does not simply add an extra layer to the input image but instead applies appropriately generated weather effects to each region of the image, to objects such as the sky, roads, and trees. Third, MWTG is able to consider the semantic information. For example, the wires connecting the power and telephone poles were partly hidden under hazy weather conditions and the lane markings were covered by ice and snow under snowy conditions. However, in the case of rainy weather, the generated results were not ideal because the patterns in the rainy weather source images were not conspicuous enough for the model to learn them effectively. We will address this problem in the future by collecting more useful rainy weather image data.

Regarding the weather removal results, we can still observe accurate color transformation and realistic scene translation results, as shown in Figure 6, but MWTG occasionally fabricated inputs, generating artifacts in some cases, most notably the insertion of fake grass in the middle of the road when removing rainy weather effects. We believe this is due to limitations in the generation process and, when weather effects are very extensive, the network is unable to determine the original context without some guessing.

#### 4.3.2. Qualitative and Quantitative Evaluation Using Cityscapes Weather Data

Here, we present our qualitative results when using the Cityscape corpora and, since ground truth images without any adverse weather phenomena are included, we will then provide quantitative results also. As shown in Figure 7, MWTG demonstrated high performance when removing weather effects during the qualitative experiment. This is because, in this setting, the only difference between the adverse and clear weather domains was the weather effects. Therefore, even though the data were unpaired during training, MWTG can still determine what is hidden behind the haze, rain streaks, or snowflakes and recover the original images.

In our quantitative evaluation, we compare MWTG with the state-of-the-art single weather removal methods, but only for their specific tasks. For de-hazing, we compared the performance of our proposed MTWG method with DehazeNet [25], MSCNN [26], AODNet [27], and GridDehazeNet [51]. For de-raining, we compared MTWG with RCDNet [52], MPRNet [53], PReNet [54], and RESCAN [55]. For de-snowing, we compared it with RESCAN [55], SPANet [56], and DesnowNet [30]. Note that although MSCNN is listed in all the comparisons, it is still categorized as a single weather removal tool since it needs to be retrained for each removal task. In contrast, MWTG performs all these tasks using the same model.

PSNR and SSIM were used to compare the performance of each model when using the Cityscapes images. The tabulated results are shown in Table 1, Table 2 and Table 3.

From these results, we can see that MWTG achieved similar or better performance than the other de-hazing and de-raining methods, as measured using PSNR and SSIM. However, for the de-snowing task, MWTG was impaired by pixel resolution differences and, thus, did not achieve satisfactory performance. This is because the conventional methods were evaluated using the original Snow Cityscapes images, with an image resolution of 512×256, while we resized these images to match the resolutions of the other two datasets (2048×1024) when training MWTG. Therefore, MWTG is dealing with images that are 16 times larger than the conventional methods.

### 4.4. Evaluation Using Perception Algorithm

To verify the suitability of MWTG for visual applications, we tested it using the state-of-the-art pedestrian detector ACSP on Foggy Cityscapes images. An example of MWTG’s de-hazing performance is shown in Figure 8 and ACSP detection results before and after de-hazing are shown in Table 4. As we can see, the detection results clearly improved after the images were de-hazed using MWTG. For further investigation, we used ACSP on the validation set of Foggy Cityscapes and tabulated the values of log-average Miss Rate over False Positive Per Image (FPPI); the results are shown in Table 5. We also apply the SOTA object detector, which uses Cascade-RCNN [57] as the backbone, on Weather Cityscapes as shown in Figure 9. We can observe the performance improvement in the detection numbers in different weather conditions.

### 4.5. Discussion

Based on the results of our evaluations, we can confirm that MWTG is able to translate images of multiple types of adverse weather into clear images, since a constraint on cycle-consistency loss allows the background context to remain unchanged. We now discuss in more detail the capabilities and drawbacks of weather generation and removal using MWTG.

In the experiment with the RDSBW dataset, the number of samples for each weather condition was unbalanced. Normally, models learn better conversion rules with more training samples. However, in the case of hazy weather, with only 2000 training samples, the model was able to achieve satisfactory results. In contrast, in the case of images of rainy weather, even though the model was trained with 4000 image samples, the translation results were less accurate. This is because rain creates more complex patterns in images. For example, streaks of rain in the air are spindly, so they are difficult for the camera to capture. Furthermore, rain is often accompanied by high humidity, thus there is often fog in the background of rainy images. When encountering finer and more varied distinctions, our model tends to learn simpler representations, which is why the rainy images generated using MWTG are more similar to an intermediate output between haze and snow. We can also observe that MWTG’s weather generation performance is superior to its weather removal performance. Since the CycleGAN model [35] is good at translation tasks involving color and texture changes and, since our model is based on CycleGAN, MWTG should inherit this ability. However, even though we intuitively consider the generation and removal of weather to be two separate tasks, our model treats both as translation tasks. This means erasing noise, such as blocks of haze or flakes of snow from occluded objects, is not the primary target of the model but it is adding a layer of snow on the road or inserting a layer of haze in the distance, for example. Note that this difference in conversion performance is less obvious when using the Cityscapes data, where domain variance is minimal since the images are all synthesized using the same dataset.

## 5. Conclusions

In this work, we have explored a solution to the visibility degradation problem that intelligent vehicles encounter when operating under adverse weather conditions, which can lead to malfunctioning of the perception module. Our proposed, dual-purpose framework, called Multiple Weather Translation GAN (MWTG), is able to perform adverse weather generation and removal tasks simultaneously. In particular, we trained our image translation model using unpaired data. Three weather generators were used to create adverse weather effects on images of normal driving scenes obtained from video datasets, while a fourth clear weather generator was used to recover clear images by removing hazy, rainy, and snowy noise. To avoid translation deviation, we added a spatial feature transform layer to fuse the feature maps of the front-end network, as an information guide to the subsequent network.

A qualitative evaluation of MWTG using our own RDSBW dataset and qualitative and quantitative evaluations using reorganized images from the Cityscapes and Cityscapes weather datasets showed that MWTG can achieve promising de-noising performance. Moreover, the results of a practical experiment showed that our model boosted the performance of state-of-the-art pedestrian detector ACSP when tested using the Foggy Cityscapes images.

In the future, we intend to expand the range of adverse weather that the model can handle to include strong light and nighttime scenes, without complicating the present MWTG framework.

## Figures and Tables

**Figure 1 sensors-23-01548-f001:**
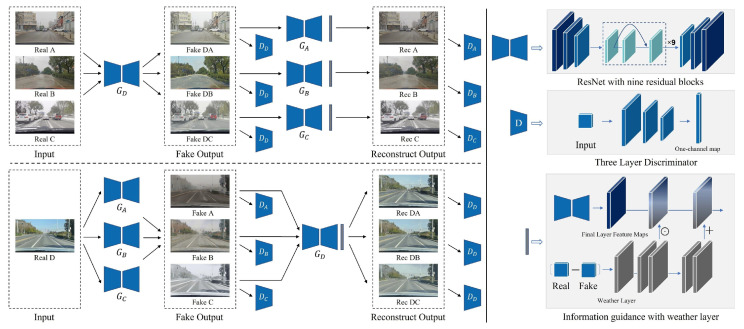
Architecture of proposed Multiple Weather Translation GAN. MWTG consists of four generators corresponding to four mappings: $Y\to x_{1}$, $Y\to x_{2}$, $Y\to x_{3}$, and X→Y. All the generators have a ResNet encoder–decoder with nine residual blocks. Their associated discriminators are a three-layer CNN and output a one-channel prediction map. The weather layers, which are obtained by subtracting the fake output from the input, are used as information guidance to achieve better results. Note that A∼D represent hazy, rainy, snowy, and clear weather, respectively, and that $G_{A∼D}$ are the respective networks used for generating these weather effects.

**Figure 2 sensors-23-01548-f002:**
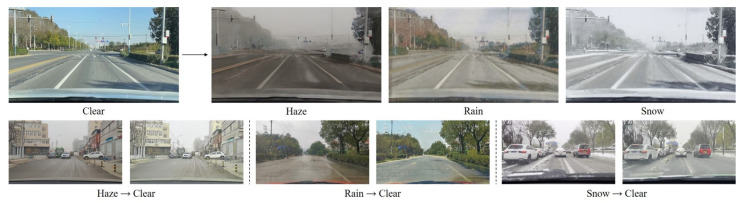
MWTG translates clear images into hazy, rainy, or snowy images using three different generators, creating a set of images representing different weather conditions (**top row**). In contrast, only one generator is needed to translate all three types of adverse weather images into clear ones (**bottom row**).

**Figure 3 sensors-23-01548-f003:**
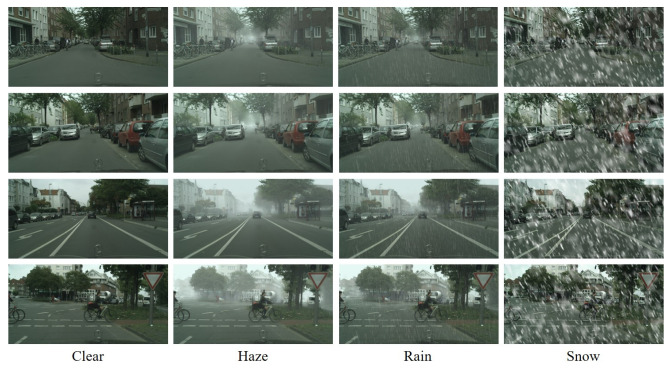
Sample scenes from the Cityscapes and rearranged Cityscapes weather datasets. We combined images selected from Foggy Cityscapes [49], Rain Cityscapes [50], Snow Cityscapes [31], and the original Cityscapes [48] datasets into one corpus consisting of 5000, 3540, 6000, and 5000 images from each dataset, respectively. The number of samples from each dataset varied due to variation in our synthesis strategy for different weather types. The original resolution of all the images is 2048×1024.

**Figure 4 sensors-23-01548-f004:**
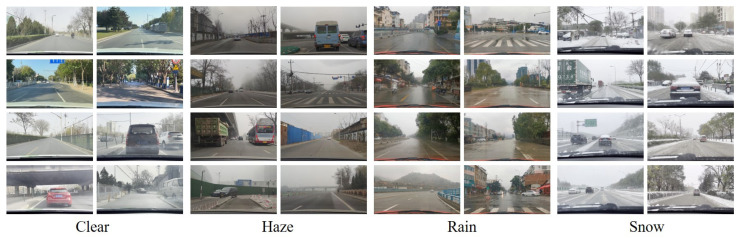
Sample scenes from the Realistic Driving Scenes under Bad Weather dataset (RDSBW). We first captured driving scene video under different weather conditions using a camera with 1920×1080 resolution mounted behind a car windshield. We then selected sharp and unique images and manually categorized them by weather type. The “haze” set contains 2052 images, the “rain” set contains 4171 images, the “snow” set contains 4777 images, and the “clear” set contains 2831 images. Note that the images are uncorrelated by location.

**Figure 5 sensors-23-01548-f005:**
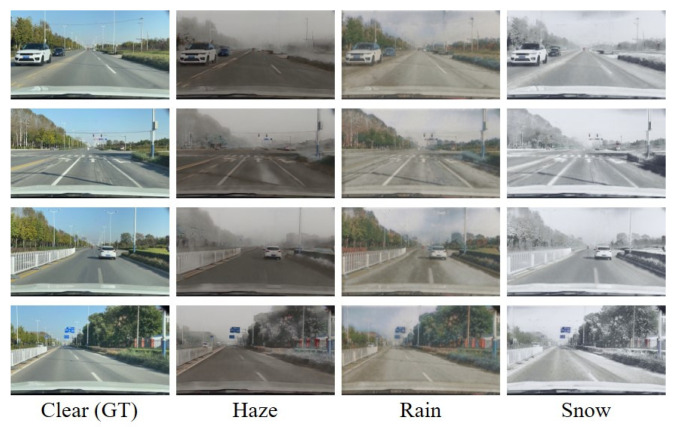
Weather generation results for RDSBW. Even when trained without paired image sets, MWTG still can translate clear images into images of our three adverse weather conditions without corrupting the background context.

**Figure 6 sensors-23-01548-f006:**
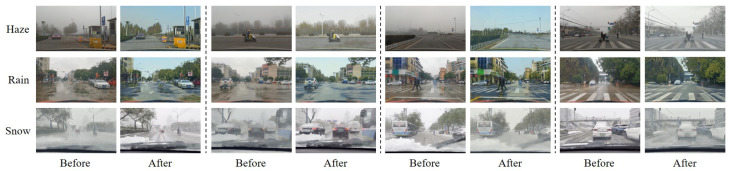
Weather removal results for the RDSBW dataset, showing examples of MWTG’s translation of adverse weather images into clear weather images; while MWTG was unable to recover objects and buildings hidden behind dense haze, it did not randomly insert fake objects.

**Figure 7 sensors-23-01548-f007:**
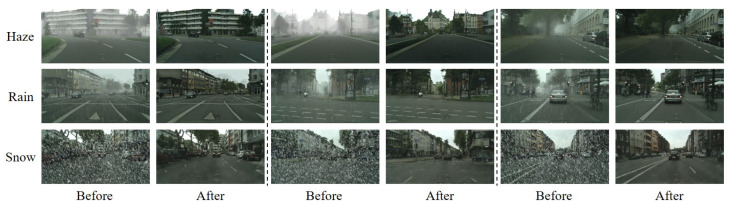
Weather removal results for the Cityscapes datasets. Although the MWTG model uses an unpaired training paradigm, the original Cityscapes dataset contains ground truth images for its synthesized weather datasets, in contrast to the RDSBW dataset. The result is that even if objects are occluded by haze, rain, or snowflakes, MWTG still can recover the original Cityscape images to generate clear images.

**Figure 8 sensors-23-01548-f008:**
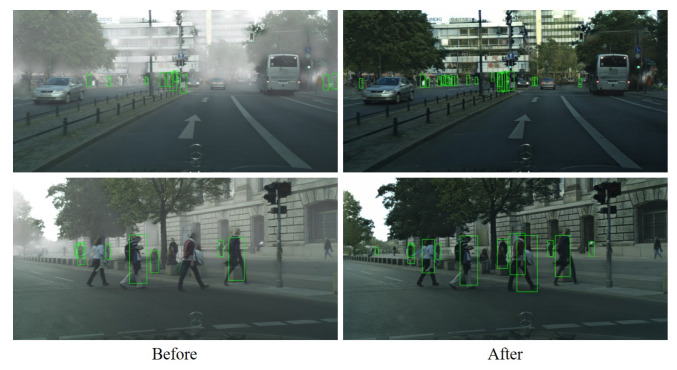
Application of MTWG on Foggy Cityscapes using SOTA pedestrian detector.

**Figure 9 sensors-23-01548-f009:**
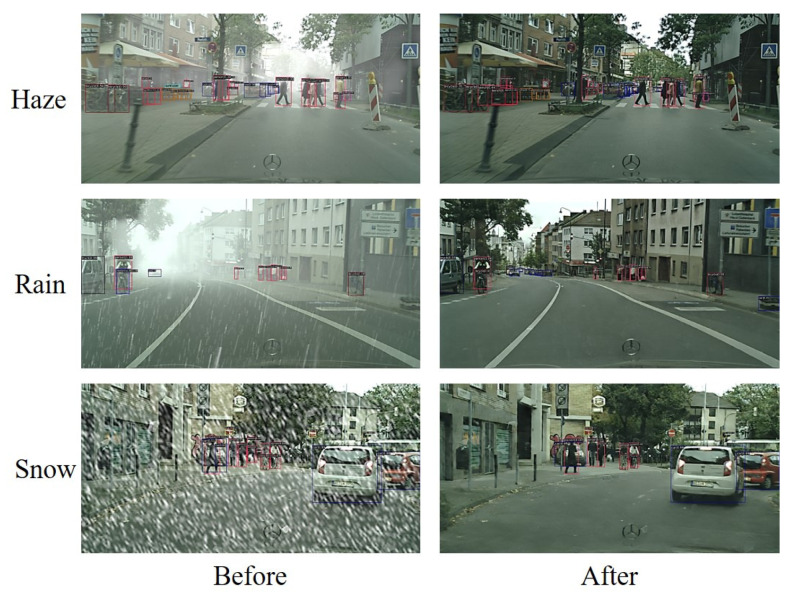
Application of MTWG on Weather Cityscapes using SOTA object detector.

**Table 1 sensors-23-01548-t001:** Comparison of results with Foggy Cityscapes.

Type	Method	Venue	PSNR	SSIM
Task Specific	DehazeNet [25]	TIP	14.971	0.487
	MSCNN [26]	ECCV	18.994	0.859
	AODNet [27]	ICCV	15.446	0.631
	GridDehazeNet [51]	ICCV	23.72	0.922
	Previous Work [24]	VTC	24.071	0.915
Multi Task	MWTG	-	23.844	0.911

**Table 2 sensors-23-01548-t002:** Comparison of results with Rain Cityscapes.

Type	Method	Venue	PSNR	SSIM
Task Specific	RCDNet [52]	CVPR	20.39	0.6498
	MPRNet [53]	CVPR	20.10	0.6815
	PReNet [54]	CVPR	20.48	0.6598
	RESCAN [55]	CVPR	20.44	0.6681
	Previous Work [24]	VTC	22.458	0.886
Multi Task	MWTG	-	25.16	0.911

**Table 3 sensors-23-01548-t003:** Comparison of results with Snow Cityscapes.

Type	Method	Venue	PSNR	SSIM
Task Specific	RESCAN [55]	ECCV	33.63	0.9627
	SPANet [56]	CVPR	35.73	0.9741
	DesnowNet [30]	TIP	33.58	0.9382
	Previous Work [24]	VTC	27.42	0.871
Multi Task	MWTG	-	25.233	0.858

**Table 4 sensors-23-01548-t004:** Generation Results on Cityscapes.

Type	Weather	PSNR	SSIM
Multi Task	Haze	21.091	0.924
	Rain	21.375	0.849
	Snow	19.021	0.679

**Table 5 sensors-23-01548-t005:** Log-average miss rate over false positive per image (FPPI) results for ACSP pedestrian detector using Foggy Cityscapes.

Data	Reasonable	Bare	Partial	Heavy
Foggy (before)	23.73%	16.32%	25.93%	58.70%
De-haze (after)	20.65%	14.98%	21.30%	58.70%

## Data Availability

Not applicable.

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
