# Peer review of "Framework for Generation and Removal of Multiple Types of Adverse Weather from Driving Scene Images"

_sensors, 2023, doi:10.3390/s23031548_

Round 1

Reviewer 1 Report

The authors proposed a framework for generation and removal of multiple types of adverse weather that has significant implications for unmanned scene perception. However, the paper needs further improvement before publication.

1. The symbols in some equations in the text are not defined, e.g. "⊙"...

2. The description of the method is very unclear and the author is advised to represent the pseudo-code.

3. The experiments are not sufficient and ablation experiments are recommended to verify the importance of each component of the proposed model.

Reviewer 2 Report

Following are my comments and suggestions:

1. Abstract: Some of the sentences are not clear. It is better to rewrite with research objective, research motivation and research usefulness.

2. Introduction: Research motivation or problems and contributions should be clearly mentioned. 

3. A flowchart of the method can improve the quality of the article.

4. Eq. 6: Please mention the base of log.

5. Eq. 7: Explanation is required.

7. There are some recent studies on quantum-based image analysis. The authors can review them as:

https://doi.org/10.1016/j.advengsoft.2022.103370

Round 2

Reviewer 2 Report

This article can be accepted.